# Circulating and Salivary NGF and BDNF Levels in SARS-CoV-2 Infection: Potential Predictor Biomarkers of COVID-19 Disease—Preliminary Data

**DOI:** 10.3390/jpm12111877

**Published:** 2022-11-09

**Authors:** Filippo Biamonte, Agnese Re, Bijorn Omar Balzamino, Gabriele Ciasca, Daniela Santucci, Cecilia Napodano, Giuseppina Nocca, Antonella Fiorita, Mariapaola Marino, Umberto Basile, Alessandra Micera, Cinzia Anna Maria Callà

**Affiliations:** 1Dipartimento di Scienze, Laboratoristiche ed Infettivologiche, UOC Chimica, Biochimica e Biologia Molecolare Clinica, Università Cattolica del Sacro Cuore, 00168 Rome, Italy; 2Laboratorio Patologia Clinica, Associazione dei Cavalieri Italiani del Sovrano Militare Ordine di Malta ACISMOM San Giovanni Battista, 00148 Rome, Italy; 3Fondazione Policlinico Universitario “A. Gemelli”, IRCCS, 00168 Rome, Italy; 4Research and Development Laboratory for Biochemical, Molecular and Cellular Applications in Ophthalmological Sciences, Research Laboratories in Ophthalmology, IRCCS-Fondazione Bietti, 00184 Rome, Italy; 5Dipartimento di Neuroscienze, Sezione di Fisica, Università Cattolica del Sacro Cuore, 00168 Rome, Italy; 6Center for Behavioural Sciences and Mental Health, Istituto Superiore di Sanità, 00161 Rome, Italy; 7Dipartimento di Scienze Biotecnologiche di Base, Cliniche Intensivologiche e Perioperatorie, Università Cattolica del Sacro Cuore, 00168 Rome, Italy; 8Dipartimento di Scienze dell’Invecchiamento, Neurologiche, Ortopediche e della Testa e del Collo, Università Cattolica del Sacro Cuore, 00168 Rome, Italy; 9Dipartimento di Medicina e Chirurgia Traslazionale, Università Cattolica del Sacro Cuore, 00168 Rome, Italy

**Keywords:** COVID-19, NGF, BDNF, SARS-CoV-2, stress, neuroinflammation, dynamic biomarkers

## Abstract

COVID-19 continues to afflict the global population, causing several pathological diseases and exacerbating co-morbidities due to SARS-CoV-2’s high mutation. Recent interest has been devoted to some neuronal manifestations and to increased levels of Nerve Growth Factor (NGF) and Brain-derived Neurotrophic Factor (BDNF) in the bloodstream during SARS-CoV-2 infection, neurotrophins that are well-known for their multifactorial actions on neuro-immune-endocrine and visual functions. Nineteen (19) patients were enrolled in this monocentric prospective study and subjected to anamnesis and biosamples collection (saliva and blood) at hospitalization (acute phase) and 6 months later (remission phase). NGF and BDNF were quantified by ELISA, and biochemical data were related to biostrumental measurements. Increased NGF and BDNF levels were quantified in saliva and serum during the acute phase of SARS-CoV-2 infection (hospitalized patients), and reduced levels were observed in the next 6 months (remission phase), never matching the baseline values. Salivary and circulating data would suggest the possibility of considering sera and saliva as useful matrices for quickly screening neurotrophins, in addition to SARS-CoV2 antigens and RNA. Overall, the findings described herein highlight the importance of NGF and BDNF as dynamic biomarkers for monitoring disease and reinforces the possibility of using saliva and sera for quick, non-invasive COVID-19 screening.

## 1. Introduction

To date (2022), coronavirus infection (COronaVIrus Disease 19; COVID-19) still remains a public health emergency continuing to affect the health of individuals and the Sanitary Services worldwide [1]. Severe Acute Respiratory Syndrome Coronavirus 2 (SARS-CoV-2) triggers respiratory tract infection, presenting mild (cold) to severe symptoms, by binding the Angiotensin-Converting Enzyme 2 (ACE2) receptor (ACE-2R) at nasopharyngeal and ocular tissues, providing direct (cell infection) and indirect (cytokine storm) effects [1,2,3,4]. Recently, the Omicron variant has overtaken the Delta one in terms of circulation/transmission, prolonging the the COVID-19 pandemic saga and the observation of long-COVID-19 states, even in vaccinated subjects [1,2,3,4,5,6,7,8,9]. One of the hardest challenges in the fight against COVID-19 is the development of wide-scale, effective, and rapid laboratory tests to control disease severity, progression, and worsening. The presence of symptoms even at follow-up highlights the importance of alternative biomarkers and possibly non-invasive laboratory tests.

Neurotrophins modulate the neuro-immuno-endocrine path, regulating various homeostatic processes and balancing parainflammation and inflamaging, working at the level of neuroimmunomodulation [10,11]. Systemic and local neurotrophin levels can change after exposure to physical, biochemical, and social stressors, regulating the psycho-neuroimmune-endocrine pathway and visual homeostasis [12,13,14,15,16,17,18,19,20,21,22,23]. Decreased Nerve Growth Factor (NGF) and Brain-Derived Neurotrophic Factor (BDNF) levels have been described in neurodegenerative diseases [10,24,25]. Circulating NGF and BDNF levels are principally ascribed to platelets and immune cells to support the survival of cholinergic neurons [13,16,21,26,27,28].

To date, only a few studies have discussed the potential role of neurotrophins in COVID-19. The neurological manifestations of COVID-19 might be the consequence of SARS-CoV-2 moving directly into the nervous system or an indirect effect of an alerted immune system (cytokine storm), impairment in the blood–brain and retinal barriers (BBB and BRB), coagulopathies, and/or complications of pre-existing illnesses (diabetes, obesity) [3,4,11,29,30,31]. The SARS-CoV-2/ACE-2R binding could impair the release of NGF and BDNF, reducing their availability for physiological functions. Liu and coworkers proposed NGF as a potential diagnostic/therapeutic target in SARS-CoV-2-induced pulmonary aggravations [15]. Recently, Azoulay and co-workers observed a correlation between the increase in circulating BDNF and the improvement in COVID-19 symptoms [32]. Fiore and coworkers proposed that serum NGF and/or BDNF levels could be useful for predicting a worsened prognosis in SARS-CoV-2 for adolescent girls and male patients, respectively [33,34]. Little is known about the changes in NGF and BDNF in adult COVID-19 patients, and no data are available when shifting from infection (acute) to non-infectious (chronic) SARS-CoV2 disease. Finally, no comparative NGF and BDNF data are available for both matrices.

This study aims to evaluate the levels of circulating and salivary NGF and BDNF in COVID-19 patients, comparing their expression between acute and remission phases of disease. Therefore, the identification of a selected panel of biomarkers could improve the clinical management of COVID-19 patients and open a new scenario for drug treatments devoted to targeting specific pathways.

## 2. Materials and Methods

### 2.1. Ethics and Study Population

This study complied with the Ethical Principles for Medical Research Involving Human Subjects in accordance with the World Medical Association Declaration of Helsinki and was certified by the Committee of the Applicable Institution of the Fondazione Policlinico Universitario Agostino Gemelli IRCCS (FPG, Rome). The intramural ethics committee approved the study (ID: 3222, date of approval: 21 May 2020).

After informed consent acquisition, 19 patients were enrolled in this monocentric prospective study among patients admitted to the COVID-19 Unit (IRCCS-FPG). Blood and salivary specimens were simultaneously collected at hospitalization between May and July 2020. The study population consisted of hospitalized COVID-19 adult patients with a positive molecular test (SARS-CoV-2) and symptoms of COVID-19 (fever > 37.5 °C, dry cough, and pharyngodynia [4,35]). Patients were followed during the remission phase at home: saliva and blood were collected after 20–30 days from symptomatology regression and after testing negative on a SARS-CoV-2 molecular test. Five (5) age- and sex-matched healthy volunteers participated in the study (controls). Exclusion criteria were young subjects (<18 years), plasma cell disorders, autoimmune diseases, autoantibodies, antinuclear antibody positivity, monoclonal component in capillaries electrophoresis, and chronic hepatitis.

### 2.2. Biosample Handling and Laboratory Tests

Samplings were performed simultaneously for each patient. Briefly, blood was collected according to standard procedures, and sera were produced after gentle centrifugation (3000 rpm/7 min). Saliva samples were collected with salivette^TM^ devices (SARSTEDT S.r.l., Milan, Italy), and swabs were centrifuged at 15,000 rpm/15 min to extract saliva from sponge [36]. Conventional DuoSet^®^ kits were used for human βNGF (DY256-05) and BDNF (DY248-05) detection (R&D Systems, McKinley Place, Minneapolis, MN, USA). Briefly, 96-well MaxiSorp™ plates (Nunc, Roskilde, Denmark) were pre-coated with the specific capture antibodies (0.4 μg/mL; 4 °C/overnight; R&D). Samples were appropriately diluted (1:2 for NGF and 1:3 for BDNF) in assay diluent (R&D) supplemented with 1× protease inhibitor cocktail (Pierce-Thermo Fisher Scientific Inc., Waltham, MA, USA) and loaded in precoated plates in parallel with the standard curve (0.32–2.000 pg/mL protein; R&D). Subsequently, the specific secondary antibodies (0.15 μg/mL; R&D) and streptavidin (1:200; R&D) were added, and the specific binding was developed by using the ready-to-use 3,3′,5,5′-Tetramethylbenzidine (TMB) substrate solution (R&D). The colorimetric signals (Optic Density, OD) were acquired at λ 450–570 nm by using a 96-well plate reader (spectrophotometer; Tecan Group Ltd., Männedorf, Switzerland). Target values were normalized for total protein (A280; Nanodrop analysis) and were produced using a 3rd grade polynomial standard curve, as calculated by Prism 9.3 (GraphPad Software Inc., San Diego, CA, USA). Concentrations of analytes are shown as pg/mL. The absence of cross reactivity with other neurotrophins was assured by the manufacturers. The analysis was performed by an operator without knowledge of the clinical information of the handled sample.

### 2.3. Statistical Analysis: Packages and Visual Representation

The data analyses were performed using the open source RStudio (RStudio|Open source and professional software for data science teams—RStudio; ver. 4.2.1, released on 23 June 2022; RStudio, PBC, Boston, MA, USA) [37,38], while the data visualization was carried out with Origin Pro 2022 (OriginLab—Origin and OriginPro—Data Analysis and Graphing Software; OriginLab Corporation, Northampton, MA, USA). Demographic and clinical characteristics of patients were summarized using the gtsummary R package [36]. Lab parameters and their time-dependent differences were tested for normality using a visual inspection of the QQ plot, followed by a Shapiro–Wilk test. Since significant deviations from normality were detected in selected variables and in the corresponding time-dependent differences, a non-parametric analysis was selected. Continuous variables were reported in terms of their medians and interquartile ranges (IQR). Qualitative variables were reported as counts or percentages. Comparisons between the biomarkers’ levels were performed using a Paired Wilcoxon Signed Rank Test and displayed in SARS-CoV-2 matrices between acute phase and follow-up. Paired data were visualized with a violin plot analysis using Origin Pro 2022. As data appear to be right-skewed, the distribution of numerical data was visualized using the best-fitted log-normal distribution. Data points measured on each patient are also reported in the analysis. Different subjects were highlighted in different colors, and the two time-points were connected with a dashed line. Correlations between variables were evaluated with Spearman’s correlation coefficients, showing both ρ and *p* Values. For visualization purposes, ρ values were arranged in a correlation matrix, according to a standard representation. No significant correlations were highlighted with an “x” symbol. A double color scale is used to visualize the strength of the correlations, with blue dots indicating the presence of negative correlations and red ones indicating positive correlations. Additionally, the larger the dot, the stronger the correlation. The correlation matrix was computed and visualized with Origin Pro 2022. A scatter plot of the data leading to significant and non-trivial correlations was also reported. Data in scatterplots were analyzed with a linear regression model. Model applicability was tested by looking at the distribution of fitting residuals. No statistically significant deviation from the normality of residuals was detected. The best regression line and equation were superimposed on each plot together with the corresponding confidence and prediction bands. Statistical significance was set as follows: ns, *p* > 0.05; *, *p* ≤ 0.05; **, *p* ≤ 0.01; ***, *p* ≤ 0.001; ****, *p* ≤ 0.0001.

## 3. Results

The demographic and clinical characteristics of the 19 patients recruited in the study are reported in Table 1. Briefly, our study population comprised patients (age range: 48–72 years; sex-matched) hospitalized from 19 to a maximum of 42 days, with (9/19) or without comorbidities (10/19), such as obesity, diabetes, and hypertension.

At the time of hospitalization, patients displayed classical symptoms of SARS-CoV-2 infection and showed comorbidities typical of hospitalized COVID-19 patients, including pneumonia (15/19), although none of the patients showed severe lung involvement (COPD) [4,15]. Six out of nineteen patients were admitted at the Intensive Care Unit, and others received oxygen therapy (9/19) and/or biological therapy (9/19) [35]. All patients were in thromboprophylaxis with enoxaparin [39].

Saliva and blood samples were collected at two different time-points: first during hospitalization (acute phase) and then 6 months later (likewise in the remission phase at home). Not all samples were available for biochemical quantifications. Biochemical data and outputs from statistical analysis of NGF and BDNF quantifications are summarized in Appendix A (see column N).

NGF and BDNF values in acute and remission phases of COVID-19 patients were first compared with those assessed in the six controls. Particularly, circulating and salivary levels of NGF increased in COVID-19 with respect to baseline (120.00 ± 70.00 pg/mL and 95.20 ± 25.60 pg/mL, respectively, in healthy controls). The same results were obtained for circulating and salivary levels of BDNF values (1050.00 ± 960.00 pg/mL and 21.81 ±  2.90 pg/mL in healthy controls). NGF and BDNF levels remain high, even in the remission phase of disease.

### 3.1. COVID-19 and NGF Levels

The two consecutive samplings were compared by the non-parametric paired Wilcoxon Signed Rank Test, superimposed for each assay, and the violin plot analyses of NGF levels in saliva (A) and serum (B) are reported in Figure 1.

Particularly, salivary NGF levels were significantly higher in the acute phase compared to remission (797.82 (725) pg/mL versus 120.03 (439) pg/mL; median (IQR) values; *p* = 0.0012; Figure 1A). On the contrary, no significant changes were detected in circulating NGF levels between the first (acute phase) and second (after 6 months) samplings (4509 (606) pg/mL versus 4640 (571) pg/mL; median (IQR) values; *p* = 0.43; Figure 1B).

Examining the data’s distribution shown by the violin plots, we observed that 3 out of 15 samples showed an unchanged or increase trend of salivary NGF levels. Of note, only 2 out of 19 samples showed a decrease in circulating NGF levels between the first and second assays. These trends are visible in the violin distribution, as different subjects are highlighted in different colors (colored dots) and the two paired measures are connected with a dashed line.

### 3.2. COVID-19 and Changes in Salivary and Circulating Levels of BDNF

The same non-parametric statistical analysis was carried out for BDNF. Salivary BDNF levels were significantly decreased in remission phase (second assay) with respect to the acute phase (first assay), as shown by the violin schematization (177 (170) pg/mL versus 44 (28) pg/mL; median (IQR) values; *p* < 0.001; *p* = 0.00085; acute versus remission phase; Figure 2A). Although a decreasing trend was observed for circulating BDNF levels between the first and the second sampling, no significant changes were provided by the statistical analysis (2377 (714) pg/mL versus 1989 (480) pg/mL; median (IQR) values; *p* = 0.25; Figure 2B).

Examining the BDNF data’s distribution offered by the violin plots, we observed that just one sample (outliner) showed an inverse expression of BDNF with respect to overall data (outliner in second assay), and 3 out of 15 samples showed unchanged salivary BDNF levels. These trends are visible in the violin distribution, as different subjects are highlighted in different colors (colored dots) and the two paired measures relate to a dashed line.

### 3.3. Demographic and Biochemical Comparisons

Therefore, we investigate the correlation between the measured laboratory markers and selected clinical and demographical variables, as shown in Figure 3. Output correlations were arranged in a correlation matrix of Spearman’s ρ coefficient (Figure 3A). Significant positive correlations were observed between age and days of hospitalization (ρ = 0.68, *p* < 0.001; *p* = 0.0013) and between salivary NGF and BDNF values, both in the first (ρ = 0.95, *p* < 0.001; *p* = 1.7 × 10^−7^) and second (ρ = 0.97, *p* < 0.0001; *p* = 1.4 × 10^−8^) assays. Trivial correlations were also observed between NGF and BDNF in the acute phase and the corresponding differences between the two consecutive samplings: NGF versus NGF difference in saliva (ρ = 0.85, *p* < 0.0001; *p* = 6.9 × 10^−5^), NGF versus NGF difference in serum (ρ = 0.73, *p* = 0.025), BDNF versus BDNF difference in saliva (ρ = 0.90, *p* < 0.0001; *p* = 2.4 × 10^−4^), and BDNF versus BDNF difference in serum (ρ = 0.93, *p* = 0.01). These latter correlations are considered trivial as they are likely to arise from the mathematical definition of the variable and will be not analyzed further. For comprehensiveness, we show a scatter plot analysis of the data underlying non-trivial correlations, namely, age and days of hospitalization (Figure 3B) and salivary NGF and BDNF in acute phase (Figure 3C) and during the follow-up (Figure 3D). Data were analyzed with a linear regression model, and the best regression line together with confidence and prediction bands are superimposed on each scatterplot. Furthermore, the linear function that provides the best fit of the data is reported above each plot, together with the statistical significance of the retrieved fitting coefficients. Qualitative observation of the analyzed scatterplots shows the presence of randomly distribute residual values around the best-fitted line, thus confirming the applicability of the regression model.

## 4. Discussion

Herein, we report the increased NGF and BDNF levels in saliva and serum collected during the acute phase of SARS-CoV-2 infection (hospitalized patients) followed by decreased levels in the next 6 months (during the remission phase). The decreased levels in the remission phase did not reach the baseline values.

Our results are supported by some old and recent studies showing: i. the participation of NGF and BDNF in viral infections, chronic lung inflammation, and tissue remodeling (fibrosis)-associated states [14,15,22,40,41,42,43,44,45]; ii. the contribution of NGF and BDNF in stress-related physio-pathological and emotional conditions [18,19,20,23,46]; and iii. the alteration of circulating or salivary NGF and BDNF levels in COVID-19 patients [32,34,47,48].

The observation of increased NGF and BDNF levels in sera from hospitalized SARS-CoV-2 positive patients might find an explanation in the activation of circulating immune cells during the inflammatory process and the “cytokine storm” associated with SARS-CoV-2 infection [31,36,49]. The cellular sources of NGF and BDNF in the bloodstream are mainly white cells (monocytes, leucocytes, and lymphocytes) and platelets, either in resting or activated states, as both neurotrophins drive autocrine and paracrine routes for survival and/or differentiation and the sustenance of cholinergic neurons [16,26,27,28]. The presence of NGF and BDNF in the bloodstream is not new, and these neurotrophins have been reported to participate in innate immune and adaptive responses and complement activation, as well as in platelet aggregation [13,26,50]. In other systems, the high circulating and tissue NGF levels upon viral infection were associated with a neuroprotective function, such as in viral conjunctivitis or in keratoplasty when Herpes Simplex reinfections might be induced upon surgical stress [40,41]. Specific to SARS-CoV-2 infection, a double-effect explanation can be proposed comprising: i. tropism toward neuronal cells; ii. specific ACE2/SARS-CoV-2 binding followed by alteration to the blood–brain barrier (BBB) and the release of soluble mediators; iii. cholinergic activation; or simply iv. a sustaining effect for immune cells [31]. Therefore, the increased levels of NGF and BDNF might be viewed as an outcome of the immune response to viral infection [22,49].

Some pilot studies on diagnostic devises highlighted the possibility to detect biomarkers in matrices other than sera [47]. In a recent study, saliva, blood and tears appeared promising matrices for the diagnosis/prognosis of diseases, at least for NGF and BDNF quantification [18]. Although blood still represents the conventional matrix for detection and/or monitoring biomarkers, its sampling could be an invasive procedure [33,48]. Saliva might offer a non-invasive alternative, in which salivary data can mirror circulating data, offering a quick, easy, and even “do-it-yourself” tool in some cases [36,51]. Herein, saliva and sera were collected and biochemically compared in order to prospect salivary tests as alternative choices to blood collection, and in case of repeated analysis, likewise those possibly performed outside sanitary services [52,53].

The high intragroup variance observed is a common feature for neurotrophins, as previously reported in other study populations with wide age ranges and of varying sizes [50]. This high variance has been reported not only for age and sex variables but also for endocrinological parameters, as well as for drug treatments and psychiatric states [50]. Herein, the presence of outliners and discordant trends between first and second assays, as visible by colored dots, might be due to the individuals’ backgrounds, which represents a limit of this study that will be future explored.

Previous studies strengthen the hypothesis that NGF and BDNF might play a functional role in stress-coping responses, as these neurotrophins promote the growth and differentiation of developing neurons in central and peripheral nervous systems, as well as the survival of neuronal cells in response to stress [33,43,44]. In the past, Aloe and coworkers demonstrated that young soldiers experiencing their first parachute jump showed a huge release of NGF into the bloodstream, probably as a result of immune cells’ activation, suggesting an association with some homeostatic adaptive mechanisms, similar to those associated with anxiety-related behaviors [50]. Of interest, the increase in NGF levels preceded the increase in plasma cortisol and adrenocorticotropic hormone, and no changes in IL1β and TNFα were found, suggesting that this increase in NGF occurred independently via inflammatory cytokines [23]. Herein, the stressful experience due to lockdown, eventual hospitalization, and even long-COVID-19 might corroborate the high levels of neurotrophins detected during the acute phase and the reduced levels at the follow-up [34,52]. It is undoubtful that some cytokines involved in “cytokine storm”, including interferon-gamma (IFNγ), some interleukins (IL-1β, IL-6, IL-2), and a few chemokines and free oxygen radicals, are well-known stimulators of the release of NGF and BDNF into the bloodstream and tissues [50].

The deposition of the extracellular matrix and the proliferation of fibroblasts in the alveolar space, key features of lung disorders, have been described in many patients who died of COVID-19 [2]. A recent study highlighted the correlation between bloodstream BDNF and the outcome of SARS-CoV-2 disease [48]. Herein, 79% of patients had pneumonia, 47% of patients required O_2_, and 100% of patients were in treatment with enoxaparin for clot prevention and reduction in cytokine storm [54]. These data do not explain the unchanged BDNF levels between the first and second assays but might suggest some other implications for BDNF.

Limitations of the study include the small population size and the lack of quantification of immature forms of neurotrophins (pro-NTs) as well as the specific tyrosine kinase (trkA and trkB) and pan-neurotrophin p75^NTR^ receptors that bind NGF and BDNF either alone or in their coupled high-affinity form [55]. A recent study reported the βNGF/trkA^NGFR^ signaling associated with anti-NP specific antibody production in mild SARS-CoV-2 infection [49]. Both NGF and BDNF bind their respective trkA^NGFR^ and trkB^BDNFR^ specific receptors in single or coupled forms depending on physiological state of the responsive cell playing inflammatory and anti-inflammatory effects on neuro-immune-endocrine and visual systems and protecting homeostasis at cardiovascular, endocrine, immune, and visual levels [42,43,44,45]. The role of their soluble receptors has been investigated, suggesting alternative mechanisms to modulate receptor activity [56]. This apparent contradiction between harmful and protective routes can be explained by considering NGF as part of a proper mechanism devoted to activating protective immune responses and dampening overt inflammatory ones, limiting tissue damage [45].

In an effort to unravel other primary potential COVID-19 biomarkers, this pilot study provides additional information aimed at disclosing further biomolecular events consequent to SARS-CoV-2 infection [36,57]. In a case report, Petrella and coworkers introduced the possibility to use serum NGF and BDNF as new predictive biomarkers of COVID-19’s long-term effects, especially in girls [34]. Herein, the regression studies might be of great interest for predicting the days of hospitalization and might be of use relative to non-vaccinated subjects. On the other hand, our findings, by showing a high correlation between NGF and BDNF in both matrices, would suggest that at least one test and one parameter, the quick-and-easy one, depending on the situation, might be useful for the prediction and/or prognosis of COVID-19. It is of note that correlation values are extremely strong when the second assay shows lower values.

## 5. Conclusions

Although the study has a limitation in its sample size, it represents a potential utility for monitoring local innate immunity.

Our study was devoted to the quantification of NGF and BDNF in COVID-19 patients, their transition from acute to remission phases, and the possibility to identify in this expression some predictors for the remission and course of COVID-19.

Saliva might be more useful and non-invasive than the blood/sera matrix. The recent interest in tear matrices has been supported by several pilot studies, highlighting an easy and non-invasive practice that does not require specific training [52].

To our knowledge, studies with comparable approaches do not exist, except for those carried out on young subjects [34]. Overall, COVID-19 infection elevates circulating and salivary NGF and BDNF levels, and even when decreasing at follow-up, the values remain high with respect to the baseline. The results of this study can be useful in prospecting assumptions for the transition from acute to chronic COVID-19, at least for NGF and BDNF, and alternative candidates for the prognosis of therapy. Additional clinical studies must still be performed to fully evaluate the ability of these mediators and their matrices as options to successfully identify COVID-19 states and perhaps predict long-COVID-19. Due to the high mutation and the appearance of different, more aggressive variants associated with different clinical signs and symptoms, it is necessary to refresh some aspects in the war against COVID-19 [58].

## Figures and Tables

**Figure 1 jpm-12-01877-f001:**
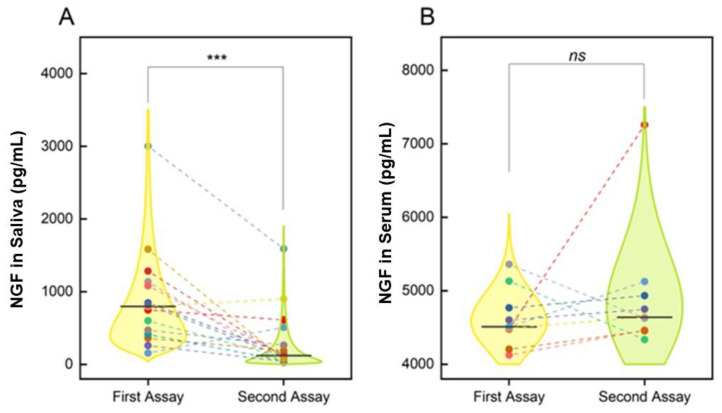
NGF changes. (**A**,**B**) Violin plot analyses for the comparison of the NGF levels in the SARS-CoV-2 acute phase and the follow-up measurement (6 months after the acute phase) in saliva (**A**) and serum (**B**) samples. Data distribution is visualized using the log-normal distribution that best fits the data. Different subjects are highlighted in different colors (dots), and paired measures are connected with dashed lines. Median values are reported as continuous horizontal lines. Non-parametric paired Wilcoxon Signed Rank Test. ***, *p* < 0.001.

**Figure 2 jpm-12-01877-f002:**
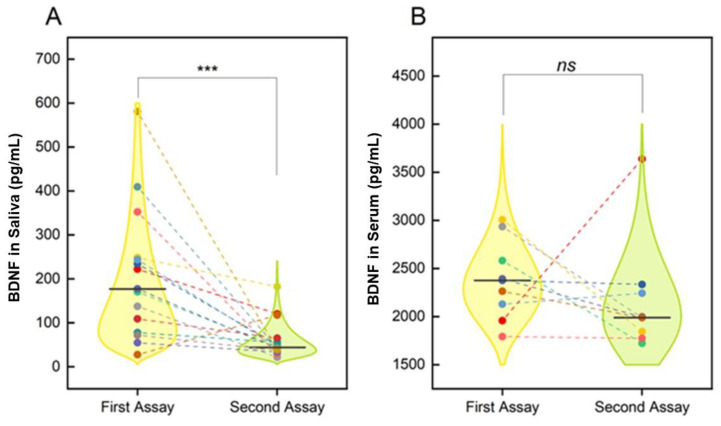
BDNF changes. (**A**,**B**) Violin plot analyses for the comparison of the BDNF levels in the SARS-CoV-2 acute phase and the follow-up measurement (6 months after the acute phase) in saliva (**A**) and serum (**B**) samples. Data distribution is visualized using the log-normal distribution that best fits the data. Different subjects are highlighted in different colors. and the two paired measures are connected with a dashed line. Median values are reported as continuous horizontal lines. Non-parametric paired Wilcoxon Signed Rank Test (***: *p* < 0.001).

**Figure 3 jpm-12-01877-f003:**
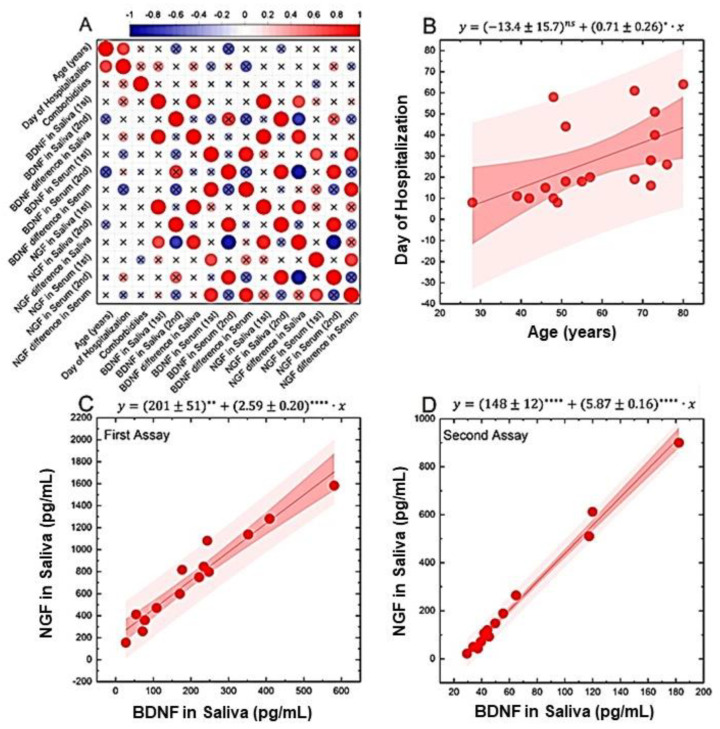
Comparisons between demographic and biochemical data. (**A**) Matrix map of selected laboratory and demographical variables. Correlations between variables were analyzed with Spearman’s correlation coefficients, and non-significant correlations are highlighted with an “x” symbol. A double color scale is used to visualize the strength and the direction of correlations: blue dots indicate the presence of negative correlations, while red dots point to positive correlations. Finally, larger dots highlight stronger correlations. (**B**–**D**) Scatter plot analysis of data leading to selected significant correlations: (**B**) days of hospitalization vs. age; (**B**) NGF vs. BDNF salivary levels in acute phase; and (**C**) NGF vs. BDNF salivary levels in remission phase. Data in scatter plots were analyzed using a linear regression model. The best regression line is superimposed on each plot together with confidence and prediction bands. Additionally, the best-fitted equation is reported on each scatter plot with the statistical significance of each retrieved fitting parameter (ns: *p* > 0.05; *: *p* ≤ 0.05; **: *p* ≤ 0.01; ****: *p* ≤ 0.0001).

**Table 1 jpm-12-01877-t001:** Study population. Demographic sum-up, comorbidities, and treatments.

Variables	Characteristics	% ^1^
Gender	F/M	8/11 (42%/58%)
Age	Years	55 (48, 72)
Hospitalization	Days	19 (13, 42)
COVID-19	Fever at onset	16 (84%)
	Dyspnea at onset	8 (42%)
	Asthenia at onset	6 (32%)
	Pneumonia	15 (79%)
Comorbidities	0	9 (47%)
	1	6 (32%)
	>1	4 (21%)
	Obesity	6 (32%)
	Hypertension	5 (26%)
	Diabetes	1 (5.3%)
Life styling	Smoke	2 (11%)
Therapies	Oxygen	9 (47%)
	Intensive Care	6 (32%)
Treatments	Enoxaparin	19 (100%)
	Anti-IL-6	9 (47%)

^1^*n* (%); Median (IQR).

## Data Availability

All data are reported in the manuscript.

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
