# Peer review of "Circulating and Salivary NGF and BDNF Levels in SARS-CoV-2 Infection: Potential Predictor Biomarkers of COVID-19 Disease—Preliminary Data"

_jpm, 2022, doi:10.3390/jpm12111877_

Round 1
Reviewer 1 Report
1. In the abstract, the background is too long. Modify the background more simply, and add methods and results in detail. In addition, change "pandemia" to "pandemic". (page 1, line 44)
2. If you focus on biomarkers and COVID-19 remission or (biomarkers and COVID-19 prognosis), describe the title, introduction, discussion, and conclusion accordingly. "Potential Predictors’ Candidate Biomarkers of COVID-19 Disease Progression" seems to be inappropriate.
3. Introduction (page 2 lines 66-67): You mentioned that little is known about the transition from acute to chronic disease in adult patients with COVID-19. However, previous studies have reported the pathophysiology of long COVID-19 or post-COVID-19 syndrome. Modify this sentence.
**Reference: Yong SJ. Long COVID or post-COVID-19 syndrome: putative pathophysiology, risk factors, and treatments. Infect Dis (Lond). 2021;53(10):737-754. doi:10.1080/23744235.2021.1924397
Higgins V, Sohaei D, Diamandis EP, Prassas I. COVID-19: from an acute to chronic disease? Potential long-term health consequences. Crit Rev Clin Lab Sci. 2021;58(5):297-310. doi:10.1080/10408363.2020.1860895
4. What is the time point for collecting blood and saliva? Did you collect blood and saliva simultaneously? If you did, this is the first study that explored circulating and salivary NGF and BDNF levels simultaneously in hospitalized COVID-19 patients. Describe more clearly the purpose, time point for collecting samples, and strengths of this study. Why is N of salivary and sera NGF/BDNF different? Was it not possible to collect consecutive sera and saliva from all 19 patients?
5. Describe the full name before using the abbreviation for the first time in the entire draft.
6. Table 2 and Figure 1 &2 present the same data. Using figures seems to be more clear. In figure 1 & 2, change "First assay" to "Acute phase", and "Second assay" to "Remission phase or 6-month follow-up". In addition, describe the term whatever you use (remission phase or 6-month follow-up) in methods.
7. The decimal places are not consistent. Display numbers excluding P value to 1 decimal place, and display numbers P value to 3 decimal places.
Ex) Table 2: P 8.54E-04--><0.001
8. Check the numbers in the tables match the numbers in the result. ex) 797.82 (725) pg/mL ---> 797.8 (411.7 - 1137.5) pg/mL
What does 725 mean? In the entire draft, numbers (IQR) are not correct.
9. In figure 3, a correlation between age and hospital days seems to be unnecessary. Correlations between biomarkers and demographic data (age, hospital days, pneumonia, comorbidities, therapy, and treatment) or differences in biomarkers according to demographic data would be more helpful.
Author Response
ANSWERS’ to reviewers
Reviewer 1
- In the abstract, the background is too long. Modify the background more simply, and add methods and results in detail. In addition, change "pandemia" to "pandemic". (page 1, line 44)
Reply All corrections were done, including abststract refining and word substitution
- If you focus on biomarkers and COVID-19 remission or (biomarkers and COVID-19 prognosis), describe the title, introduction, discussion, and conclusion accordingly. "Potential Predictors’ Candidate Biomarkers of COVID-19 Disease Progression" seems to be inappropriate.
Reply. According to request, we removed the word “candidate” from title and text. We strengthen in the discussion the usefulness of NGF and BDNF as quantified in salivary and circulating matrix as specific biomarkers of COVID-19 progression, and not prognosis nor remission of disease.
- Introduction (page 2 lines 66-67): You mentioned that little is known about the transition from acute to chronic disease in adult patients with COVID-19. However, previous studies have reported the pathophysiology of long COVID-19 or post-COVID-19 syndrome. Modify this sentence.
Reply we apologize for this improper description, according to request the sentence “Little is known about the transition from acute to chronic disease in adult patients, and no data are available for NGF and BDNF in both matrices.” was implemented as follows “Little is known about the NGF and BDNF changes in adult COVID-19 patients, and no data are available when shifting from infection (acute) to noninfectious (chronic) SARS-CoV2 disease. Finally, no comparative NGF and BDNF data are available for both matrices.”.
**Reference:
Yong SJ. Long COVID or post-COVID-19 syndrome: putative pathophysiology, risk factors, and treatments. Infect Dis (Lond). 2021;53(10):737-754. doi:10.1080/23744235.2021.1924397
Higgins V, Sohaei D, Diamandis EP, Prassas I. COVID-19: from an acute to chronic disease? Potential long-term health consequences. Crit Rev Clin Lab Sci. 2021;58(5):297-310. doi:10.1080/10408363.2020.1860895
Reply: We thank for the references suggested by the reviewers, that we added to the text. The references were added to the reference list as ref.8 (Yong et al., 2021) and ref.9 (Higgins et al., 2021), appropriately edited and all references were renumbered in the text.
- What is the time point for collecting blood and saliva? Did you collect blood and saliva simultaneously? If you did, this is the first study that explored circulating and salivary NGF and BDNF levels simultaneously in hospitalized COVID-19 patients. Describe more clearly the purpose, time point for collecting samples, and strengths of this study. Why is N of salivary and sera NGF/BDNF different? Was it not possible to collect consecutive sera and saliva from all 19 patients?
Reply: Sampling occurred simultaneously for both saliva and venous blood. The sampling occurred when the patients were first hospitalized and after 20-30 days from symptomatology regression and negative SARS-CoV-2 PCR test. The referee is right, this is the first study investigating the simultaneous expression of these neurotrophins in both matrices. To specify, the phrase “Patients were followed during the remission phase at home: saliva and blood were collected after 20-30 days from symptomatology regression and after testing negative for SARS-CoV-2 molecular test.” was added.
- Describe the full name before using the abbreviation for the first time in the entire draft.
Reply the appropriate revision was carried out. Briefly, COronaVIrus Disease 19 (COVID-19); Severe Acute Respiratory Syndrome CoronaVirus 2 (SARS-CoV-2); Angiotensin-Converting Enzyme 2 (ACE2); Angiotensin-Converting Enzyme 2 (ACE2) receptor (ACE-2R); Nerve Growth Factor (NGF); Brain-Derived Neurotrophic Factor (BDNF); 3,3',5,5'-Tetramethylbenzidine (TMB); Interferon-gamma (IFNγ); Blood Brain Barrier (BBB); Meanwhile some typing errors were fixed.
- Table 2 and Figure 1 &2 present the same data. Using figures seems to be more clear. In figure 1 & 2, change "First assay" to "Acute phase", and "Second assay" to "Remission phase or 6-month follow-up". In addition, describe the term whatever you use (remission phase or 6-month follow-up) in methods.
Reply The referee is right, we prefer to keep the figure and we can move the Table2 as supplementary table S1.
- The decimal places are not consistent. Display numbers excluding Pvalue to 1 decimal place, and display numbers P value to 3 decimal places. Ex) Table 2: P54E-04--><0.001
Reply The referee is right, all corrections were carried out and the statistical paragraph was implemented.
- Check the numbers in the tables match the numbers in the result. ex) 797.82 (725) pg/mL ---> 797.8 (411.7 - 1137.5) pg/mL What does 725 mean? In the entire draft, numbers (IQR) are not correct.
Reply In the result paragraph, we provided all values as median (interquartile range, IQR) arbitrary Unit. Therefore, a phrase was introduced in the statistical paragraph to mention the “median (IQR) values”. Since the panels were Violin plot analysis, the IQR, the difference between Q3 and Q1, would better describe the range of the middle 50% of the data.
- In figure 3, a correlation between age and hospital days seems to be unnecessary. Correlations between biomarkers and demographic data (age, hospital days, pneumonia, comorbidities, therapy, and treatment) or differences in biomarkers according to demographic data would be more helpful.
Reply The referee is right, for this reason we showed in the panels the strong correlation between NGF and BDNF in the saliva samples at hospitalization (C) and after remission (D). Panel B was added to display the characteristic of this study population.
Reviewer 2 Report
This study evaluated levels of circulating and salivary NGF and BDNF in COVID-19 patients comparing their expression between acute and remission phases of disease in order to identify the biomarker panel to improve the clinical management of COVID-19 patients. The obtained results showed potential utility of investigated biomarkers for monitoring local innate immunity.
The main question addressed by the research is the potential use of two biomarkers NGF and BDNF for the improvement of clinical management of COVID-19 patients.
This topic is original and relevant as the possibility for new drug treatments. Studies with comparable approaches do not exist, except for those carried out on young subjects.
Paper is well written and the text is clear and easy to read. The conclusions are consistent with the evidence and arguments presented. The results address the main questions posed.
The only suggestion is to revise the abstract and compose it according to instructions for authors.
Author Response
We thank reviewer for considering positively our manuscript. We accurately revised the abstract section as suggested.
Round 2
Reviewer 1 Report
The authors have addressed all my concerns, and I recommend this version for publishing.